# A non-Hermitian optical atomic mirror

Yi-Cheng Wang [1,2] ✉, Jhih-Shih You [3] ✉ & H. H. Jen [2] ✉

Explorations of symmetry and topology have led to important breakthroughs in quantum optics, but much richer behaviors arise from the non-Hermitian nature of light-matter interactions. A high-reflectivity, non-Hermitian optical mirror can be realized by a two-dimensional subwavelength array of neutral atoms near the cooperative resonance associated with the collective dipole modes. Here we show that exceptional points develop from a nondefective degeneracy by lowering the crystal symmetry of a square atomic lattice, and dispersive bulk Fermi arcs that originate from exceptional points are truncated by the light cone. From its nontrivial energy spectra topology, we demonstrate that the geometry-dependent non-Hermitian skin effect emerges in a ribbon geometry. Furthermore, skin modes localized at a boundary show a scale-free behavior that stems from the long-range interaction and whose mechanism goes beyond the framework of non-Bloch band theory. Our work opens the door to the study of the interplay among non-Hermiticity, topology, and long-range interaction.

The exquisite control of light-matter interactions is centrally important to construct new quantum optical setups and attain new functionalities. Recent research has shown that under the control of the cooperative response of dipole modes, an atomic array with subwavelength spacings can be characterized as a high-reflectivity optical atomic mirror[1,2]. A number of interesting predictions for this optical mirror include enhanced photon storage[3], topological quantum optics[4,5], quantum information processing[6], and coherent perfect absorption[7]. These phenomena can be identified from the band structures of collective atomic excitations, where the quasimomenta modes inside and outside the light cone exhibit distinct behaviors, respectively[1,3]. Specifically, the former delineate non-Hermitian physics. Due to the intrinsically loss processes associated with free-space emission, the atomic array with photon-mediated dipole-dipole interactions opens the door to the observation of a wide range of outstanding non-Hermitian phenomena that would be challenging in condensed matter.

Recent progress in non-Hermitian physics[8,9] reveals two phenomena that have no Hermitian counterparts. One is the exceptional points (EPs)[10], at which both the complex eigenvalues and eigenstates of a non-Hermitian matrix coalesce. The EPs can emerge from the implementation of non-Hermiticity in a Hermitian system with Dirac or Weyl points, which have been demonstrated in diverse physical systems, including photonic crystals[11–13], topolectrical circuits[14], and exciton-polariton systems[15,16]. The Riemann surface topologies associated with these EPs results in the nontrivial winding in the complex energy plane that induces the other intriguing non-Hermitian phenomenon—non-Hermitian skin effect[17–19].

The non-Hermitian skin effect (NHSE)[20–22] means that an extensive number of exponentially localized eigenstates can pile at the boundaries under open boundary conditions (OBCs). This indicates the breakdown of the conventional bulk-boundary correspondence[22], which has triggered an avalanche of research aimed at reestablishing the correspondence in non-Hermitian systems[23]. Various complementary approaches have been proposed in this vein, including the celebrated non-Bloch band theory[20,24,25]. This band theory successfully interprets NHSE and the exponential localization of skin modes in one-dimensional (1D) systems with finite-range couplings. Yet, little is known about the interplay between the long-range interaction and these non-Hermitian phenomena.

In this work, we consider two-dimensional (2D) atomic lattices with resonant dipole-dipole interactions (RDDIs)[26,27], which serve as a high-reflectivity optical mirror at cooperative resonance[1,2]. To efficiently calculate the bulk band structures of infinite 2D atomic lattices, we develop a model-independent generalization of Euler-Maclaurin formula[28] (see Methods). Our numerical method is applicable to systems with different types of long-range interactions. Due to the inherent

[1]Department of Physics, National Taiwan University, Taipei 10617, Taiwan. [2]Institute of Atomic and Molecular Sciences, Academia Sinica, Taipei 10617, Taiwan. [3]Department of Physics, National Taiwan Normal University, Taipei 11677, Taiwan. ✉e-mail: r09222006@ntu.edu.tw; jhihshihyou@ntnu.edu.tw; sappyjen@gmail.com

non-Hermiticity of dipole-dipole interaction, there is no Hermitian limit of our system, such that EPs should emerge from the mechanism different from splitting Hermitian degeneracy points by adding non-Hermiticity. Here we demonstrate that paired EPs can be split from a symmetry-protected nondefective degeneracy point (NDP) by a symmetry-breaking perturbation. With the nontrivial winding in the complex energy plane arising from EPs, we find that a ribbon geometry exhibits extensive geometry-dependent skin modes. In particular, these modes show a scale-free behavior that stems from the long-range interaction and the mechanism responsible for this behavior goes beyond the framework of non-Bloch band theory. Furthermore, we show that the skin modes can emerge in 2D finite atomic arrays by manipulating the orientations of open boundaries and the lattice configurations. Possible experimental observations are also discussed.

## Results

### Non-Hermitian degeneracy points

We consider a 2D rectangular atomic lattice spanned by two direct lattice vectors $\mathbf{a}_1 = a/\eta \mathbf{e}_x$ and $\mathbf{a}_2 = a\mathbf{e}_y$ with a lattice constant ratio $|\mathbf{a}_2|/|\mathbf{a}_1| = \eta$ in free space (Fig. 1a, b). Each atom has a V-type energy level composed of one ground state $|g\rangle$ and two circularly-polarized excited states $|\pm\rangle = \mp(|x\rangle \pm i|y\rangle)/\sqrt{2}$ such that the system supports two in-plane polarizations with an atomic transition wavelength $\lambda$ and decay rate $\Gamma_0$. In the circularly-polarized basis, the non-Hermitian dynamics of a single excitation is described by the following two-band effective Hamiltonian kernel[29,30] (Supplementary Notes 1 and 3)

$$\mathcal{H}_{\mathrm{eff}}(\mathbf{k}) = \hbar(\omega_0 + \Omega_{\mathbf{k}})\begin{pmatrix} 1 & 0 \\ 0 & 1 \end{pmatrix} + \hbar\Gamma_0 \begin{pmatrix} 0 & \kappa_{+-}(\mathbf{k}) \\ \kappa_{-+}(\mathbf{k}) & 0 \end{pmatrix}, \quad (1)$$

where $\mathbf{k}$ is the Bloch momentum in the irreducible Brillouin zone and $\omega_0 = 2\pi c/\lambda$ is the atomic transition frequency. Here Eq. (1) has the identical momentum-dependent interacting energy $\hbar\Omega_{\mathbf{k}}$ for $|\pm\rangle$, and $\kappa_{+-(-+)}(\mathbf{k})$ describes the couplings between two circularly-polarized states. According to the expression of RDDI in the reciprocal space (Supplementary Note 1), the 2D Brillouin zone exhibits two kinds of distinct collective excitations separated by the light cone $|\mathbf{k}| = 2\pi/\lambda$[1] (Fig. 1a, b), wherein the dissipative modes couple to far-field radiation, while the modes with $|\mathbf{k}| > 2\pi/\lambda$ related to evanescent wave confined to the atomic lattice plane are dissipationless. Here, we focus on the non-Hermitian physics within the light cone.

The bulk eigenenergy spectrum can be obtained as $E_{1,2}(\mathbf{k}) = \hbar(\omega_0 + \Omega_{\mathbf{k}}) \pm \hbar\Gamma_0\sqrt{\kappa_{+-}(\mathbf{k})\kappa_{-+}(\mathbf{k})}$, and the existence of degeneracy corresponds to $\kappa_{+-}(\mathbf{k})\kappa_{-+}(\mathbf{k}) = 0$. If $\kappa_{+-}(\mathbf{k})$ and $\kappa_{-+}(\mathbf{k})$ are simultaneously zero, the corresponding eigenstates are linearly independent such that $\mathcal{H}_{\mathrm{eff}}(\mathbf{k})$ is diagonalizable, and these degeneracies are called nondefective degeneracy points. If only one of $\kappa_{+-(-+)}(\mathbf{k})$ is zero, $\mathcal{H}_{\mathrm{eff}}(\mathbf{k})$ is nondiagonalizable, and the defective degeneracies whose eigenstates coalesce are known as exceptional points. Therefore, for a square lattice, $\kappa_{+-(-+)}(\mathbf{k})$ vanish at high symmetry point $\Gamma$ ($\mathbf{k} = 0$) due to the $C_4$ rotational symmetry (Supplementary Note 3), which ensures a symmetry-protected NDP. Accordingly, EPs can emerge by breaking this symmetry, and we note that ref. 31 has proved that NDP in a two-band system is unstable, i.e., it can be deformed into EPs by a generic perturbation.

We perform a 2D generalization of the Euler-Maclaurin formula to determine the photonic band structures of infinite square (Fig. 1c) and rectangular (Fig. 1d) lattices with in-plane polarizations in freespace. Here we diagonalize the effective Hamiltonian as $\mathcal{H}_{\mathrm{eff}}(\mathbf{k}) = V(\mathbf{k})E(\mathbf{k})V^{-1}(\mathbf{k})$, where $E(\mathbf{k})$ is the diagonal matrix composed

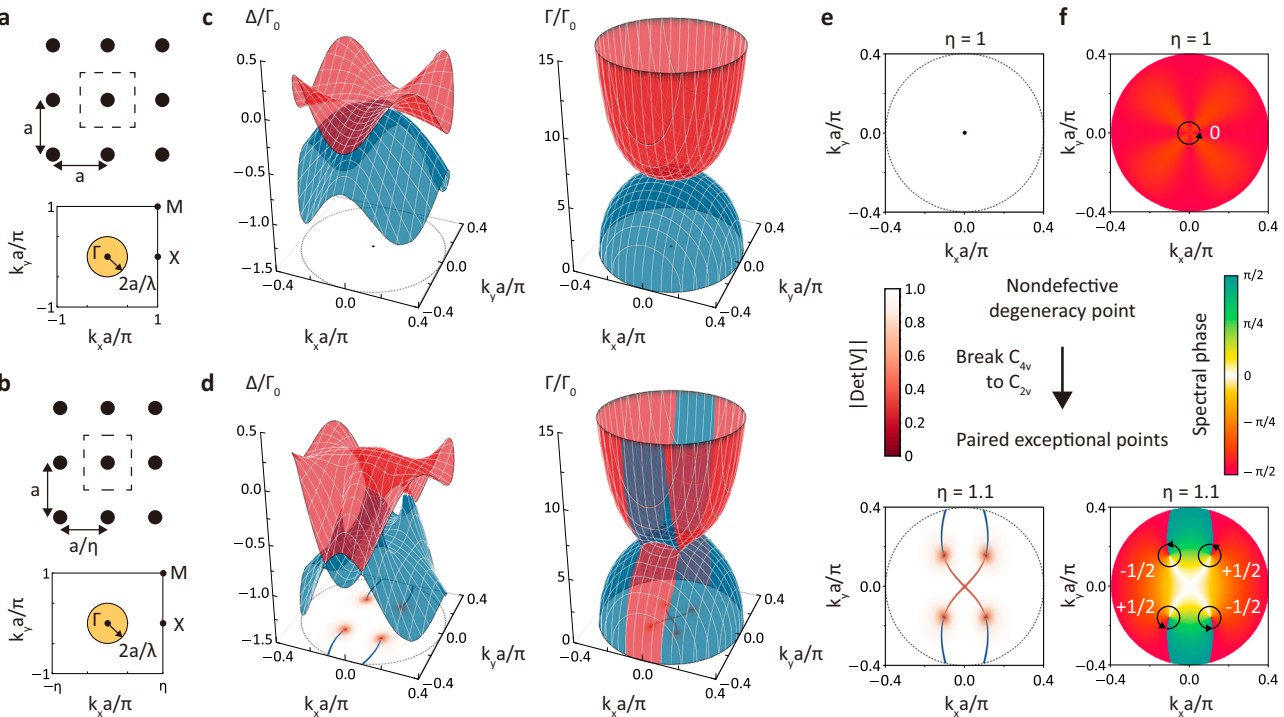

**Fig. 1 | Paired exceptional points split from a nondefective degeneracy point.** Schematics of square (**a**) and rectangular (**b**) atomic lattices and their irreducible Brillouin zones in the subwavelength regime. The yellow circular region represents the light cone, wherein the system is non-Hermitian. Collective frequency shift $\Delta_{\mathbf{k}}$ and overall decay rate $\Gamma_{\mathbf{k}}$ of infinite square (**c**) and rectangular (**d**) lattices within the light cone (black dashed circle). Two energy bands $E_{1,2}(\mathbf{k}) = \hbar(\omega_0 + \Delta_{\mathbf{k}}) - \frac{i}{2}\hbar\Gamma_{\mathbf{k}}$ are colored in red and blue, respectively. **e** A non-Hermitian degeneracy point can be identified as NDP or EP by calculating $\det[V(\mathbf{k})]$. The $\eta = 1$ case shows that the non-Hermitian degeneracy point at the high symmetry point $\Gamma$ in **c** is an NDP, and the $\eta = 1.1$ case shows that four non-Hermitian degeneracy points in (**d**) corresponding to the coalescence of two eigenstates are EPs. These EPs are joined by the degeneracy of the real and imaginary parts of $E_{1,2}(\mathbf{k})$ in (**d**), known as real (blue) and imaginary (red) Fermi arcs in the $k_x$-$k_y$ plane. **f** The spectral phase that reflects the winding of bulk energy bands. The vorticity of NDP at $\eta = 1$ is zero, while that of each EP at $\eta = 1.1$ is a half-integer. The plots are obtained with a subwavelength lattice constant $a = 0.2\lambda$ and $\eta = 1.1$ for rectangular lattice.

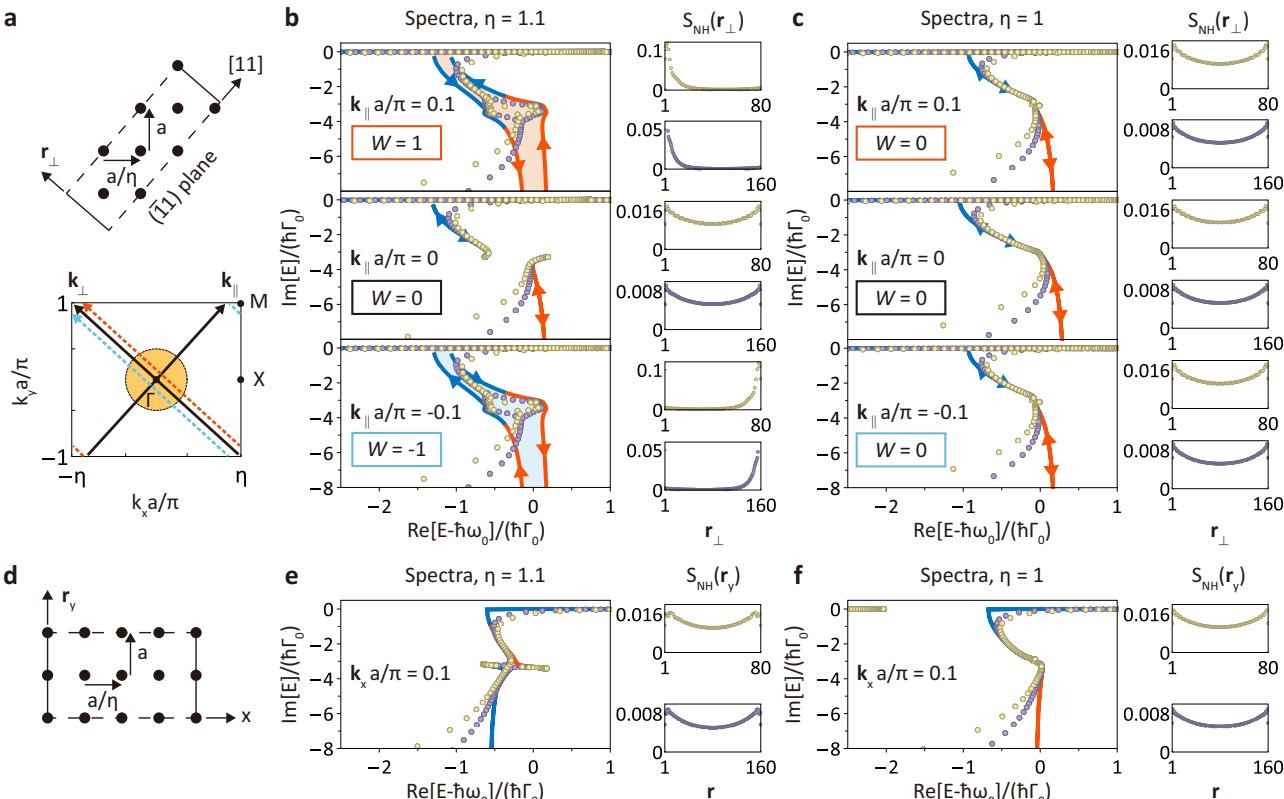

**Fig. 2 | Geometry-dependent non-Hermitian skin effect in a ribbon geometry. a** Illustration of the ribbon geometry for a rectangular atomic lattice. The boundaries are open on the ($\bar{1}1$) plane (dashed line) and extend infinitely in the [11] direction (solid line). Open boundary eigenenergy spectra $\sigma[\mathcal{H}_L(\mathbf{k}_\parallel, \mathbf{r}_\perp)]$ of rectangular (**b**, $\eta = 1.1$) and square (**c**, $\eta = 1$) lattices in ribbon geometries with a width of 80 and 160 unit cells (light yellow dots and purple dots, respectively) and the corresponding bulk spectra $\sigma[\mathcal{H}_{\text{eff}}(\mathbf{k}_\parallel, \mathbf{k}_\perp)] = \{E_{1,2}(\mathbf{k}_\parallel, \mathbf{k}_\perp)\}$ (curves in red and blue) at fixed $\mathbf{k}_\parallel = 0.1\pi/a$ (orange dashed line in **a**) and $\mathbf{k}_\parallel = 0$ ($\mathbf{k}_\perp$ axis in **a**) and $\mathbf{k}_\parallel = -0.1\pi/a$ (cyan dashed line in (**a**)). The non-Hermitian parts of $\sigma[\mathcal{H}_{\text{eff}}(\mathbf{k}_\parallel, \mathbf{k}_\perp)]$ result in the

nontrivial winding (in (**b**) at finite $\mathbf{k}_\parallel$), and the corresponding spatial distributions demonstrate that there are extensive skin modes localized at the edge normal to $\mathbf{r}_\perp$ axis in a ribbon geometry. Otherwise, inversion and mirror symmetries lead to doubly degenerate spectral arcs with zero winding numbers, which suppress the NHSE. **d** Illustration of the ribbon geometry for a rectangular atomic lattice with open boundaries in the $y$ direction. Open boundary eigenenergy spectra $\sigma[\mathcal{H}_L(\mathbf{k}_x, \mathbf{r}_y)]$ of rectangular (**e**, $\eta = 1.1$) and square (**f**, $\eta = 1$) lattices in ribbon geometries. The mirror symmetries here lead to doubly degenerate spectral arcs and suppress the NHSE. The plots are obtained with the same parameters in Fig. 1.

of its eigenvalues $E_{1,2}(\mathbf{k})$ and $V(\mathbf{k})$ is formed by two normalized right eigenstates. The coalescence of eigenstates happens when $\det[V(\mathbf{k})]$ approaches zero in the vicinity of EPs. The degeneracy point at $\Gamma$ in Fig. 1c is identified as a NDP due to the corresponding nonzero $\det[V(\mathbf{k})]$ in Fig. 1e at $\eta = 1$. In Fig. 1d, we break the $C_4$ rotational symmetry by tuning the lattice constant ratio ($\eta \neq 1$), and we find that four EPs are split from the NDP. We stress that the mechanism of the EPs here is essentially different from that in ref. 7 since the EPs here lie in the 2D Brillouin zone and arise from the lowering of crystalline symmetry.

In Fig. 1e at $\eta = 1.1$, we observe that these four EPs are joined by dispersive bulk Fermi arcs[12], along which the real parts of two eigenenergies are degenerate. We note that our bulk Fermi arcs do not lie on the isofrequency surface due to the momentum dependent $\hbar\Omega_\mathbf{k}$. These open-end bulk Fermi arcs usually terminate at EPs; however, they are truncated by the light cone in our case. Importantly, these dispersive bulk Fermi arcs and EPs are topologically stable and associated with a non-Hermitian topological invariant called the vorticity[32–34]

$$\nu = -\oint_C \frac{d\mathbf{k}}{2\pi} \cdot \nabla_\mathbf{k} \arg[E_1(\mathbf{k}) - E_2(\mathbf{k})], \qquad (2)$$

where $C$ is a counterclockwise closed loop that encloses a degeneracy point. The vorticity can be determined by the spectral phase $\arg[E_1(\mathbf{k}) - E_2(\mathbf{k})]$ that acquires a $\pm\pi$ change around each EP but 0 around NDP. In Fig. 1f, we find that the vorticity of NDP (EPs) in a square

(rectangular) lattice is zero (half-integer). We also check the stabilities of these non-Hermitian degeneracy points via the Zeeman splitting arising from a magnetic field. We find that EPs persist, in stark contrast to the symmetry-protected NDP and Dirac point (Supplementary Fig. 9).

## Non-Hermitian skin effect

Recently, it was revealed that a 1D system under OBC exhibits NHSE as long as the energy spectrum of the corresponding bulk Hamiltonian $\mathcal{H}_{1D}(k)$ encloses a nonzero spectral area in the complex energy plane. If a system has the reciprocity and mirror symmetry, $\mathcal{H}_{1D}(k) = \mathcal{H}_{1D}^T(-k)$, the bulk spectrum forms a doubly degenerate spectral arc. As a result, there is no NHSE under OBC for a 1D system with reciprocal couplings, but NHSE can emerge by invoking nonreciprocal couplings to break such a constraint on $\mathcal{H}_{1D}(k)$. Since RDDI is reciprocal, a 1D atomic chain with RDDI does not exhibit NHSE. Therefore we turn to a 2D atomic lattice with reciprocal and anisotropic RDDIs to investigate NHSE.

NHSE in 1D systems has a clear picture, while a general description of NHSE in two and higher-dimensional systems remains unclear. To study NHSE in 2D atomic lattices, we start by considering a ribbon geometry, which extends infinitely along the parallel direction (the [11] direction in Fig. 2a) and has OBC with $L$ unit cells in the perpendicular direction. Due to the translational symmetry in the parallel direction, the Bloch momentum $\mathbf{k}_\parallel$ is a good quantum number, such that the Hamiltonian of the ribbon geometry at a fixed $\mathbf{k}_\parallel$ reduces to the Hamiltonian kernel $\mathcal{H}_L(\mathbf{k}_\parallel, \mathbf{r}_\perp)$ of an effective 1D finite lattice under

OBC (over the $(\bar{1}1)$ plane in Fig. 2a). Consequently, the NHSE can be understood in a simple effective 1D picture.

In Fig. 2b, c, we numerically calculate the respective OBC spectra $\sigma[\mathcal{H}_L(\mathbf{k}_\parallel, \mathbf{r}_\perp)]$ of rectangular and square lattices in ribbon geometries with a width of $L$ unit cells at $\mathbf{k}_\parallel = \pm 0.1\pi/a$ and $\mathbf{k}_\parallel = 0$. For these given $\mathbf{k}_\parallel$, the corresponding bulk spectra $\sigma[\mathcal{H}_{\text{eff}}(\mathbf{k}_\parallel, \mathbf{k}_\perp)] = \{E_{1,2}(\mathbf{k}_\parallel, \mathbf{k}_\perp)\}$ are shown for comparison and can be viewed as the spectra of infinite 1D lattices with the normal momentum $\mathbf{k}_\perp$ being a good quantum number. First we note a deviation between the OBC spectra and the bulk spectra in the vicinity of the light cone: the bulk spectra show a divergent behavior, which arises from the RDDI in the infinite system, while the OBC spectra remain finite.

The emergence of NHSE in ribbon geometries is manifest in the bulk spectra of infinite 1D lattices. In the square lattice with oblique open boundaries (Fig. 2c) and both the rectangular and square lattices with horizontal open boundaries (Fig. 2e, f), the bulk spectra become doubly degenerate spectral arcs due to the mirror symmetry[35], indicating that the OBC eigenstates are delocalized. In the rectangular lattice with oblique open boundaries, we note that OBC eigenstates can also be delocalized at $\mathbf{k}_\parallel = 0$. This is due to the inversion symmetry at this special momentum. In the absence of these lattice symmetries, the bulk spectra enclose nonzero spectral areas, as shown in Fig. 2b at finite $\mathbf{k}_\parallel$. This signifies the hallmark of geometry-dependent skin modes.

To visualize NHSE, we consider the average spatial distribution of OBC normalized right eigenstates $|\psi_n^R(\mathbf{r}_\perp)\rangle$

$$S_{\text{NH}}(\mathbf{r}_\perp) = \frac{1}{N'} \sum_{n=1}^{N'} \sum_{j=\pm} |\psi_{nj}^R(\mathbf{r}_\perp)|^2, \qquad (3)$$

where $\pm$ represents the in-plane polarization and $n$ is in the ascending order of the imaginary parts of eigenenergies. Here we only consider first $N'$ right eigenstates $|\psi_n^R(\mathbf{r}_\perp)\rangle$ with largest decay rates. In Fig. 2b, c, $N'/2L$ corresponds to the fraction of bulk eigenmodes within the light cone at a given $\mathbf{k}_\parallel$ and we show the average spatial distribution $S_{\text{NH}}(\mathbf{r}_\perp)$ for both $L = 80$ and $L = 160$. It is evident that for a rectangular lattice in a ribbon geometry with a positive (negative) $\mathbf{k}_\parallel$, $S_{\text{NH}}(\mathbf{r}_\perp)$ shows the emergence of extensive skin modes localized at the $\mathbf{r}_\perp = 1$ ($\mathbf{r}_\perp = L$) boundary. For other cases, the OBC eigenstates are delocalized.

As a topological phenomenon, the emergence of extensive skin modes is predicted by the point-gap topology of the bulk bands[18]. For a fixed $\mathbf{k}_\parallel$ and a reference point $E_r$ that is not covered by the bulk spectra (point-gap) in the complex energy plane, the integer-valued spectral winding number can be defined as[18,31]

$$W(\mathbf{k}_\parallel, E_r) = \oint_{C_\perp} \frac{d\mathbf{k}_\perp}{2\pi i} \cdot \nabla_{\mathbf{k}_\perp} \log \det[\mathcal{H}_{\text{eff}}(\mathbf{k}_\parallel, \mathbf{k}_\perp) - E_r \mathbb{I}], \qquad (4)$$

where $C_\perp$ forms a closed loop at a fixed $\mathbf{k}_\parallel$ in the 2D Brillouin zone (e.g., three paths in Fig. 2a). When the bulk spectra enclose a nonzero spectral area in the complex energy plane, $W(\mathbf{k}_\parallel, E_r)$ for $E_r$ within the spectral area is nonzero, where we denote the corresponding spectral winding numbers by the shaded regions in Fig. 2b (+1 for orange and −1 for cyan shaded regions, respectively). Accordingly, the skin modes that lie within the interiors of bulk spectra localize at the left or right boundaries when $W(\mathbf{k}_\parallel, E_r) = \pm 1$, which coincides with other systems with finite-range couplings[18] even if we have long-range interactions here. In addition, by tuning the lattice constant ratio $\eta$ in Fig. 2b, c at a fixed, finite $\mathbf{k}_\parallel$, a point-gap opening in the ribbon geometry at nonzero $\mathbf{k}_\parallel$ represents a non-Hermitian topological phase transition[34]. This is the direct consequence of exceptional points of bulk Hamiltonian, which results in non-trivial winding in the complex energy plane and non-Hermitian skin effect in a rectangular-lattice ribbon geometry.

We note that, although the divergent bulk spectrum shows up near the light cone in Fig. 2b, c, the corresponding spectral winding numbers are quantized (Supplementary Note 4) and can be applied to characterize the point-gap topology. In addition, this NHSE is sensitive to the orientation of the open boundary. For instance, both the square and rectangular lattices can support NHSE when the OBC is imposed on the $(\bar{1}2)$ plane in a ribbon geometry since there is no mirror symmetry along the [21] direction. In general, the effective coupling in a ribbon geometry at a fixed $\mathbf{k}_\parallel$ is nonreciprocal unless the original coupling is isotropic or there is a mirror symmetry over the open boundary[36]. Therefore, the NHSE in the ribbon geometry depends on the orientation of the open boundary and the lattice configuration. We emphasize that non-Hermiticity and anisotropy are the essential ingredients for such a geometry-dependent nonreciprocal coupling.

## Scale-free localization

In addition to the deviation between the OBC and bulk spectra, we observe the scale-free behavior of spatial distributions $L S_{\text{NH}}(\mathbf{r}_\perp, L) = L' S_{\text{NH}}(\mathbf{r}_\perp, L')$ for system size $L = 80$ and $L' = 160$ in Fig. 2b, c. This implies that the number of skin modes in a ribbon geometry and their characteristic length $\xi$ of the exponentially decreasing probability $|\psi_{nj}^R(\mathbf{r}_\perp)|^2 \sim e^{-|\mathbf{r}_\perp|/\xi}$ are proportional to the system size $L$. We note that similar scale-free behavior arises in critical NHSE[37,38] in the systems with finite-range couplings. However, the scale-free localization in the atomic array stems from the long-range interaction and the mechanism underpinning this behavior goes beyond the framework of the non-Bloch band theory.

Here we briefly introduce the non-Bloch band theory in 1D systems with finite-range couplings. By considering the analytic continuation of Bloch momentum $k \rightarrow k + i\kappa(k)$[20,24,39], the OBC energy spectrum $\{E_{\text{OBC}}\}$ in the thermodynamic limit ($L \gg 1$) can be obtained from the non-Bloch Hamiltonian $\mathcal{H}_{\text{1D}}(k + i\kappa(k))$. In addition, each OBC eigenstate of a 1D chain with lattice constant $a$, $\psi_{L\rightarrow\infty}(ma)$ at $m$th site, can be decomposed into $M$ possible $\beta_i^m = e^{i(k_i + i\kappa(k_i))\cdot ma}$ and is dominated by two non-Bloch modes $\beta_{r,s}^m$ with the same modulus $|\beta| = |\beta_r| = |\beta_s|$ that corresponds to the decay length $-a/\log|\beta|$. These $\beta_i$ are solutions to the following characteristic equation (Supplementary Note 4)

$$\det[\mathcal{H}_{\text{1D}}(\beta) - E_{\text{OBC}}\mathbb{I}] = 0 \qquad (5)$$

subject to OBCs, and $M$ is determined by the coupling range. These results are derived from the asymptotic behavior of the OBC eigenstate $\psi_L(ma)$ in the finite system for large $L$. We note that the above statement is valid when the coupling range is finite. It is because in this case the finite systems with different sizes are governed by the 'same' Hamiltonian $\mathcal{H}_{\text{1D}}(\beta)$.

In contrast, in the presence of long-range RDDI the finite systems with different sizes are governed by 'different' $\mathcal{H}_L(\beta)$, such that the characteristic equation becomes size-dependent and has $M = 4(L-1)$ solutions of $\beta$. As a result, both the OBC spectrum (Fig. 3a) and $\beta$ depend on $L$, and each OBC eigenstate in the thermodynamic limit is dominated by several $\beta$ with no fixed scale. To show the size dependence of eigenstate, we numerically fit the characteristic length $\xi$ of the probability of each OBC eigenstate in Fig. 3c, which presents the crossover from a constant to scale-free characteristic length.

## Non-Hermitian skin effect in 2D finite-size systems

We further explore NHSE in a 2D finite atomic array with parallelogram-shaped boundaries by investigating the spatial distribution of skin modes (Eq. (3)). In Fig. 4, we choose $N' = [\pi(a/\lambda)^2/\eta] \times 2L_x L_y$ that corresponds to the fraction of bulk eigenmodes within the light cone in 2D Brillouin zone. In Fig. 4d, there are extensive skin modes localized at

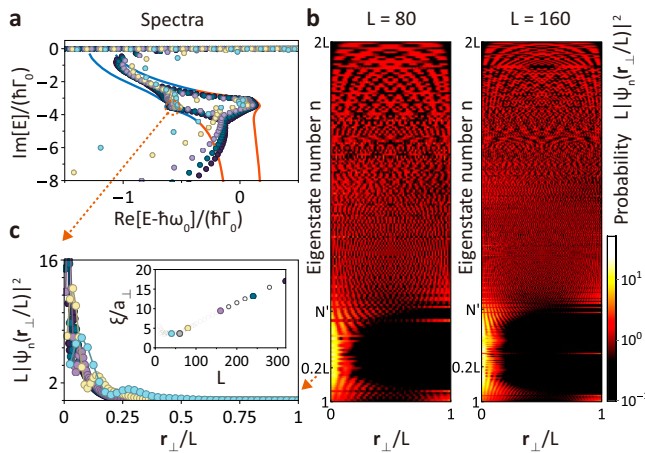

**Fig. 3 | Size dependence of non-Hermitian skin effect in a ribbon geometry.**
**a** The OBC spectra at $L = 40, 80, 160, 240,$ and $320$ unit cells (light blue, light yellow, purple, dark green, and black, respectively) gradually approach bulk spectrum as $L$ increases. **b** Rescaled probability distributions of normalized right eigenstates at 80 and 160 unit cells in the ascending order of the imaginary part of eigenenergy. $N'$ represents the number of localized modes in spatial distributions in Fig. 2b, c.
**c** Rescaled probability distributions of the $n = 0.2L$ eigenstates. The inset shows a crossover from a constant to scale-free characteristic length as system size increases, and $a_\perp$ is the lattice constant for this ribbon geometry in the $\mathbf{r}_\perp$ direction. The plots are obtained with the same parameters in Fig. 2b at $k_\parallel = 0.1\pi/a$.

the oblique ([11] direction) but not at the horizontal ([10] direction) boundaries, and there is no skin mode in Fig. 4a–c. We note that the corner accumulation of $S_{NH}(\mathbf{r})$ in Fig. 4c does not show NHSE; instead, it arises from the long-range interaction (Supplementary Fig. 6).

To understand the NHSE here, we can extend a finite system infinitely along one open boundary to reconstruct a ribbon geometry. This allows us to determine which boundary extensive skin modes are localized at by identifying its energy spectrum topology. For instance, there are two possible ribbon geometries for the finite system in Fig. 4d. One extends infinitely along the oblique direction (Fig. 2a) and the other extends infinitely along the horizontal direction (Fig. 2d). The skin modes emerge in the former case (Fig. 2b), while the NHSE in the latter case is suppressed by the mirror symmetry (Fig. 2e). Therefore, the population distributions of non-Hermitian eigenstates rely on the orientations of open boundaries and the lattice configurations. The comparison with the ribbon geometry provides a way to understand a so-called geometry-dependent NHSE[40] in two and higher dimensional systems. In the same manner, we note that the scale-free localization found in a ribbon geometry could also be observed in a 2D finite-size lattice (Supplementary Fig. 7).

In Fig. 4, the OBC spectra with the same lattice configuration (Fig. 4a, c for square lattice and b, d for rectangular lattice) are insensitive to the orientations of open boundaries. However, when the lattice configuration changes from square to rectangular lattice, the $E_{OBC}$ around NDP in Fig. 4a, c are deformed into $E_{OBC}$ around EP in Fig. 4b, d. We further explore the relationship between OBC and bulk spectra by the light scattering from this finite system (Supplementary Note 6). We only need two detunings to extract the bulk Hamiltonian from the scattering matrices in the finite system. This allows us to identify the bulk band structure and its non-Hermitian degeneracy points from light scattering.

The frequency shift and the linewidth are comparable in Fig. 4, which smears the optical response from each skin mode and its anomalous transport behavior[41]. However, when an incident light shines on a 2D atomic array that hosts skin modes at oblique incidence, the dynamical property of induced dipoles shows the asymmetric spatial distribution, which reflects the nonreciprocal effective coupling induced by the nonzero in-plane momentum of the incident light. This

provides an experimental signature of NHSE in 2D finite atomic arrays. Moreover, we could assemble two atomic lattices with distinct geometries to make skin modes localize at the interface[42], which further affects the dipole distribution.

## Discussion

We have shown that a 2D atomic array with the inherent non-Hermiticity and the anisotropy of RDDI presents nontrivial topologies that have no Hermitian counterparts. Our results here can be manifested in a general setting of coupled 2D quantum emitters[43]. Besides these distinct non-Hermitian degeneracies and NHSE we uncover here, the interplay between non-Hermiticity and topology[23,44–46] may further generate topologically protected edge states robust to not only Hermitian but also non-Hermitian defects. In addition, one can couple the atomic arrays with different electromagnetic environments to further modify the non-Hermiticity of RDDI and to enhance the optical response of some selective skin modes. Our work paves the way towards the exploration of exotic many-body states in two and higher-dimensional systems and opens up new opportunities in manipulating topological properties by tailoring long-range interactions in an atomic array.

## Methods
### 2D generalization of Euler-Maclaurin formula
In the calculation of photonic band structures, we encounter an infinite summation of RDDI $\sum_{\mathbf{R}\neq 0} e^{-i\mathbf{k}\cdot\mathbf{R}}\mathbf{G}_0(\mathbf{R})$ excluding the self-energy $\mathbf{G}_0(\mathbf{0})$, where $\mathbf{G}_0(\mathbf{R})$ is the free-space dyadic Green's function and the summation runs over all direct lattice vectors except for $\mathbf{R} = \mathbf{0}$. Since the oscillating terms result in the slow convergence, we can rewrite this infinite summation in the reciprocal space as $(\mathbf{a}_1 \times \mathbf{a}_2)^{-1}\sum_{\mathbf{G}}\mathbf{g}_0(\mathbf{G}+\mathbf{k}) - \mathbf{G}_0(\mathbf{0})$, where the summation runs over all reciprocal lattice vectors $\mathbf{G}$ and $\mathbf{g}_0(\mathbf{G}+\mathbf{k})$ is Green's function in reciprocal space. Here we note that the self interaction $\mathbf{G}_0(\mathbf{0})$ is just the integral of $\mathbf{g}_0(\mathbf{G}+\mathbf{k})$, i.e., $\mathbf{G}_0(\mathbf{0}) = \int(2\pi)^{-2}d^2G\mathbf{g}_0(\mathbf{G}+\mathbf{k})$. Therefore, an infinite summation of RDDI becomes the difference between the summation $\sum_{\mathbf{G}}$ and the integration $\int(2\pi)^{-2}(\mathbf{a}_1 \times \mathbf{a}_2)d^2G$ of the same function $(\mathbf{a}_1 \times \mathbf{a}_2)^{-1}\mathbf{g}_0(\mathbf{G}+\mathbf{k})$. While both the summation and the integration are divergent due to the self-interaction, their difference is physically meaningful and convergent. References[29,30] perform such calculations invoking the ultraviolet frequency cutoff. However, the infinite summation of RDDI $\sum_{\mathbf{R}\neq 0} e^{-i\mathbf{k}\cdot\mathbf{R}}\mathbf{G}_0(\mathbf{R})$ is independent of the frequency cutoff. By using the generalization of Euler-Maclaurin formula, we can compute this difference without requiring a frequency cutoff since it is in the form of the difference between the summation and the integration of the same function.

The original Euler-Maclaurin formula is for a single summation on a 1D lattice. The difference between the summation and the integration of the same function over any interval $I$ composed of unit cell sandwiched between lattice sites $j$ and $j+1$ can be approximated by the function and its higher-order derivatives on two ends of the interval $\partial I$. Here we extend this idea to the double summation on a 2D lattice, where a 1D interval $I$ with two ends becomes a 2D region $R$ whose boundary has a variety of shapes, such as a simply-connected region whose boundary is a closed loop and a hollow region with the inner and outer boundaries.

By means of Euler-Maclaurin formula, we can estimate this difference over any region composed of the periodic unit cell in the reciprocal space by the correction terms consisting of higher-order derivates of $(\mathbf{a}_1 \times \mathbf{a}_2)^{-1}\mathbf{g}_0(\mathbf{G}+\mathbf{k})$ at the boundaries. In avoid of the singularity arising from the light cone, the periodically-tiled hollow region $R$ we use to perform the Euler-Maclaurin formula should enclose the light cone in the reciprocal space. When the outer boundary of $R$ extends infinitely, the correction terms at this boundary become negligible due to the presence of ultraviolet frequency cutoff. Thus, the rest correction terms only lie on the inner boundary $\partial R$ of $R$, and

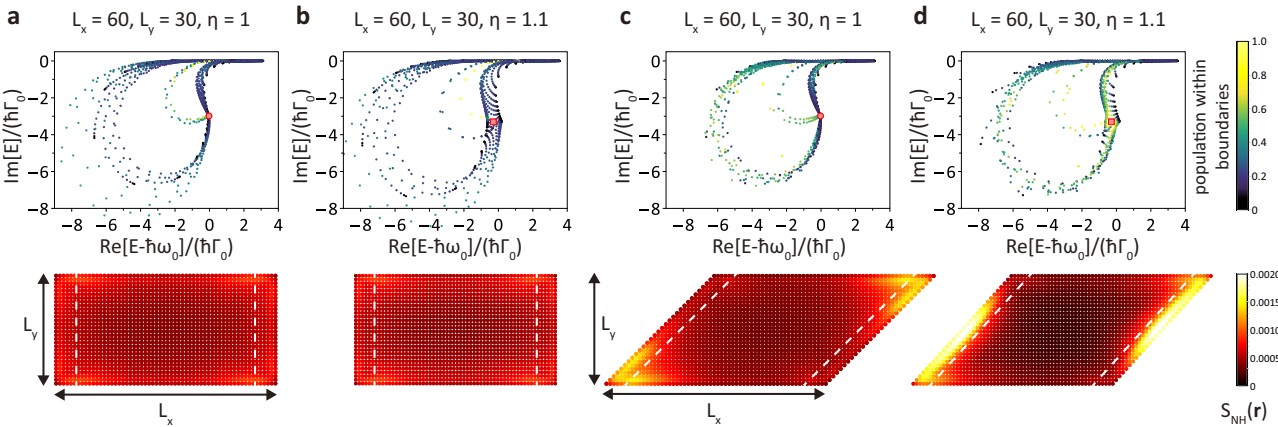

**Fig. 4 | Geometry-dependent non-Hermitian skin effect in rectangle and parallelogram-shaped boundaries.** Spectra and spatial distribution of non-Hermitian right eigenstates of 2D square (**a**, **c**) and rectangular (**b**, **d**) atomic lattices ($a = 0.2\lambda$) with rectangle (**a**, **b**) and parallelogram (**c**, **d**) shaped boundaries. At a fixed system size $L_x \times L_y = 60 \times 30$, only the geometry in (**d**) exhibits extensive skin modes localized at left and right boundaries since mirror symmetries of the other geometries suppress NHSE. Each mode is colored according to its population within the given boundaries (out of white dashed lines, leftmost and rightmost 6 sites), and those extensive skin modes in (**d**) are located around the exceptional points in the corresponding bulk spectra. In all the spectra, the nondefective degeneracy points and exceptional points are denoted by the red circles and red rectangles, respectively.

the infinite summation reduces to the finite summation and integration of $(\mathbf{a}_1 \times \mathbf{a}_2)^{-1} \mathbf{g}_0 (\mathbf{G} + \mathbf{k})$ over the finite region enclosed by $\partial R$ and several correction terms on $\partial R$ in the reciprocal space. We note that this method is applicable for the infinite summation due to other long-range interactions. Further details of the explicit formula and application are presented in the Supplementary Note 2.

## Data availability
The data in this manuscript are available from the authors upon reasonable request.

## Code availability
The code used for the analysis are available from the authors upon reasonable request.

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

## Acknowledgements
We thank Hui Liu for useful discussions. Y.-C.W. and H.H.J. acknowledge support from the Ministry of Science and Technology (MOST), Taiwan, under the Grant No. MOST-109-2112-M-001-035-MY3. Y.-C.W. and J.-S.Y. are supported by the Ministry of Science and Technology, Taiwan (Grant No. MOST-110-2112-M-003-008-MY3). J.-S.Y. and H.H.J. are also grateful for support from National Center for Theoretical Sciences in Taiwan.

## Author contributions
Y.-C.W. conducted the analytical and numerical calculations. J.-S.Y. and H.H.J. conceived the idea and supervised the project. All authors contributed to the writing of the manuscript.

## Competing interests
The authors declare no competing interests.
