## [Peer Review File · Nature Communications]

REVIEWER COMMENTS

Reviewer #1 (Remarks to the Author):

The authors Wang, You & Jen have studied two-dimensional subwavelength array of neutral atoms near the co-operative resonances of collective modes of long-range interactions. Exceptional points of the system are calculated and non-Hermitian skin effect identified. There has been considerable interest in non-Hermitian physics and exceptional points. The long-range coupling provides a novel angle on this, although there are also works on different long-range models in related systems.

This is an interesting topic and the paper is well written, but I am not convinced the work is suitable for Nature Communication. It is very technical and the consequences of the exceptional points and the skin effect are not explored. There are long calculations of the band structure that follow Perczel et al Photonic band structure of two-dimensional atomic lattices (only cited in Supplementary) and similar derivations by Antezza et al Spectrum of Light in a Quantum Fluctuating Periodic Structure Phys. Rev. Lett. 103, 123903 (not cited). Especially it would be interesting to study the skin effect further. There is a calculation of the exceptional points in the subwavelength array atomic mirror already by Ballantine et al arXiv:2012.04393 which also considers coupling between different atom transitions. The authors should compare this with their calculations.

Reviewer #2 (Remarks to the Author):

In this work, the authors investigate a two-dimensional non-Hermitian atomic lattice with long-range interaction. First of all, they show that the exceptional points appear in the momentum space within the light cone because of the dissipation nature in this region. Due to the existence of the exceptional points, since the spectral area becomes finite, the spectral winding number defined in Eq. (4) takes nonzero values, and then, the geometry-dependent non-Hermitian skin effect occurs. In fact, in the ribbon geometry, the energy spectra corresponding to the localized eigenstates surround the finite area as shown in Fig. 2. This is why we can find the skin modes only in the parallelogram geometry as shown in Fig. 4d. Furthermore, the authors propose that the scale-free localization occurs in this system because of the long-range interaction.

I think that it is interesting to propose the realization of the new type of the non-Hermitian skin effect in a realistic system because there is a lack of studies on the non-Hermitian skin effect in higher-dimensional non-Hermitian systems. Furthermore, it is useful to be able to intuitively understand the mechanism of the occurrence of the non-Hermitian skin effect through the emission within the light

cone. This is because this mechanism is applicable to other photonic systems. Thus, I believe that this work is important, and it can open new field of researches on non-Hermitian physics.

I have some questions and comments about the contents in the manuscript.

(i) Why does the energy spectrum which is the function of the wave number vector becomes discontinuous in Fig. 2? We can see in Fig. S4 that the discontinuity appears in the vicinity of the light cone. Does the system exhibit the phase transition around this region?

(ii) I think that Fig. 2 does not express the geometry-dependent non-Hermitian skin effect. This figure only shows that in the ribbon geometry which is regarded as a one-dimensional system, the conventional non-Hermitian skin effect occurs, and the energy spectrum corresponding the localized states forms the winding structure. However, Ref. [34] proposed the correspondence between the spectral area in a two-dimensional system with a full periodic boundary condition and the non-Hermitian skin effect which only occurs in a higher-dimensional system. Thus, the authors confused the discussion in a one-dimensional system with that in a two-dimensional system, and the discussion provided here is misleading. Therefore, the authors should modify this point.

(iii) I understand the mechanism of the occurrence of the scale-free localization in the ribbon geometry because of the long-range interaction. This phenomenon is interesting because it is associated with the non-Hermitian skin effect, and it has no counterpart in Hermitian systems. However, I think that it is difficult to make the ribbon geometry in the atomic-lattice system. Thus, how can we observe the scale-free localization in this system? Furthermore, does the scale-free localization occur under a full open boundary condition?

(iv) In Fig. 4c, why does the eigenstates seem to be localized at the corners? Do these eigenstates exhibit the non-Hermitian skin effect?

(v) In Figs. 4b and 4d, where are the nondefective degeneracy point and the exceptional point?

(vi) In this work, the authors only focus on the rectangle geometry and on the parallelogram geometry. I think that the non-Hermitian skin effect also occurs in other geometry. How can we quantitatively investigate the dependence on geometry of the non-Hermitian skin effect?

If the authors answer my questions and appropriately modify the manuscript, I recommend to publish this paper in Nature Communications.

Reviewer #1 (Remarks to the Author):

The authors Wang, You & Jen have studied two-dimensional subwavelength array of neutral atoms near the co-operative resonances of collective modes of long-range interactions. Exceptional points of the system are calculated and non-Hermitian skin effect identified. There has been considerable interest in non-Hermitian physics and exceptional points. The long-range coupling provides a novel angle on this, although there are also works on different long-range models in related systems.

We appreciate the Reviewer's interests in our work and valuable comments to improve our manuscript. Indeed, non-Hermitian physics has recently raised considerable interests in quantum optics and condensed matter communities. Specifically, here in an optical atomic mirror with long-range dipole-dipole interactions, we uncover the exceptional points emerged by lowering the crystal symmetry of a square atomic lattice and discover the geometry-dependent non-Hermitian skin effect (NHSE). We also present the non-Hermitian topological phase transitions, a direct consequence of exceptional points of bulk Hamiltonian, and reveal the non-trivial windings in the complex energy plane and the associated NHSE in a ribbon geometry.

We resonate with the Reviewer that the long-range couplings indeed provide a distinct angle on the non-Hermitian physics in an atomic mirror we study here. For example, our system with long-range couplings presents scale-free localization behavior which goes beyond the non-Bloch band theory often utilized in explaining NHSE in one-dimensional (1D) systems with finite range couplings.

This is an interesting topic and the paper is well written, but I am not convinced the work is suitable for Nature Communication. It is very technical and the consequences of the exceptional points and the skin effect are not explored. There are long calculations of the band structure that follow Perczel et al Photonic band structure of two-dimensional atomic lattices (only cited in Supplementary) and similar derivations by Antezza et al Spectrum of Light in a Quantum Fluctuating Periodic Structure Phys. Rev. Lett. 103, 123903 (not cited). Especially it would be interesting to study the skin effect further.

We thank the Reviewer again for appreciation of our work. We agree that we start from a similar Hamiltonian as those in Phys. Rev. A 96, 063801 (2017) and Phys. Rev. Lett. 103, 123903 (2009) since a similar light-matter interacting system with long-range dipole-dipole interaction is explored. To understand the system behavior in momentum spaces, they calculate the photonic band structure and need to invoke an ultraviolet frequency cutoff to regulate the self-interaction. However, the complex eigen-spectrum involving collective Lamb shifts and collective decay rates should not depend on the artificial frequency cutoff. Therefore, we apply the Euler-Maclaurin formula without requiring frequency cutoff to obtain the band structures, which can further allow us to extend our technique to other types of long-range interactions. The band structure calculation is straightforward, but the technique we develop here, mostly put in the supplemental material, is a more physical and to the least an alternative approach to resolve the complex eigen-spectrum of any non-Hermitian

systems with long-range interactions. In the revised version of the manuscript, we have cited these two references as Refs. [29,30] and include above discussion in Methods for clarity (underlined parts are modified from the previous version):

“While both the summation and the integration are divergent due to the self-interaction, their difference is physically meaningful and convergent. References [29,30] perform such calculations invoking the ultraviolet frequency cutoff. However, the infinite summation of RDDI $\sum_{R \neq 0} e^{-ik \cdot R} G_0(R)$ is independent of the frequency cutoff. By using the generalization of Euler-Maclaurin formula, we can compute this difference without requiring a frequency cutoff since it is in the form of the difference between the summation and the integration of the same function.”

For the consequences of the exceptional points (EPs), we note that the point-gap opening by tuning the lattice constant ratio shown in Fig. 2b, c is the direct consequence of exceptional points of the bulk Hamiltonian, which also results in the NHSE in a rectangular-lattice ribbon geometry. To clarify this point, in the “**Non-Hermitian skin effect**” section in the main text, we add “*This is the direct consequence of exceptional points of bulk Hamiltonian, which results in non-trivial winding in the complex energy plane and non-Hermitian skin effect in a rectangular-lattice ribbon geometry.*” For the consequences of the NHSE, we have mentioned that it results in the asymmetric in the dipole distribution in our main text.

We agree that it would be interesting to study the skin effect further. One aspect is the fundamental mechanism of its emergence under long-range interactions in high-dimensional systems, which we focus here and have revealed the distinct scale-free behavior in the ribbon geometry in the original manuscript. Following the suggestions by Reviewer #2, we have also performed additional investigations on the scale-free behavior in open 2D arrays in the revised manuscript. The other aspect is its applications, which are under active investigations now, and some of them involve, for example, the topological manipulations of light in Science 368, 311 (2020) (Ref. [42] in the revised main text) This point is now addressed in the “**Non-Hermitian skin effect in 2D finite-size systems**” section of the main text (underlined parts are modified from the previous version): “*...This provides an experimental signature of NHSE in 2D finite atomic arrays. Moreover, we could assemble two atomic lattices with distinct geometries to make skin modes localized at the interface [42], which further affects the dipole distribution.*” We expect new phenomena along with novel applications and more to come when topology and symmetry breaking are intertwined in non-Hermitian systems.

There is a calculation of the exceptional points in the subwavelength array atomic mirror already by Ballantine et al arXiv:2012.04393 which also considers coupling between different atom transitions. The authors should compare this with their calculations.

The exceptional points in Nanophotonics 10, 1357 (2021) by Ballantine and Ruostekoski arise by tuning the level shift between two uniform collective excitations in the finite atomic array at normal incidence, i.e. out-

of-plane and in-plane polarizations. This can be viewed as one Bloch mode at the center of the Brillouin zone as the system becomes infinity. In our work on the contrary, we study the exceptional points in the whole momentum space by lowering the crystal symmetry, and here we focus on the in-plane polarizations. Therefore, the mechanism and consequences of the exceptional point in Nanophotonics 10, 1357 (2021) are essentially different from ours.

Based on the fact that Nanophotonics 10, 1357 (2021) also discussed the non-Hermitian properties of a finite two-dimensional atomic array, we cite this work in the introduction part as Ref. [7] in the revised main text, and the comparison is now presented in the “**Non-Hermitian degeneracy points**” section in the main text:

“We stress that the mechanism of the EPs here is essentially different from that in Ref. [7] since the EPs here lie in the 2D Brillouin zone and arise from the lowering of crystalline symmetry.”

Reviewer #2 (Remarks to the Author):

In this work, the authors investigate a two-dimensional non-Hermitian atomic lattice with long-range interaction. First of all, they show that the exceptional points appear in the momentum space within the light cone because of the dissipation nature in this region. Due to the existence of the exceptional points, since the spectral area becomes finite, the spectral winding number defined in Eq. (4) takes nonzero values, and then, the geometry-dependent non-Hermitian skin effect occurs. In fact, in the ribbon geometry, the energy spectra corresponding to the localized eigenstates surround the finite area as shown in Fig. 2. This is why we can find the skin modes only in the parallelogram geometry as shown in Fig. 4d. Furthermore, the authors propose that the scale-free localization occurs in this system because of the long-range interaction.

I think that it is interesting to propose the realization of the new type of the non-Hermitian skin effect in a realistic system because there is a lack of studies on the non-Hermitian skin effect in higher-dimensional non-Hermitian systems. Furthermore, it is useful to be able to intuitively understand the mechanism of the occurrence of the non-Hermitian skin effect through the emission within the light cone. This is because this mechanism is applicable to other photonic systems. Thus, I believe that this work is important, and it can open new field of researches on non-Hermitian physics.

We thank the Reviewer for the careful reading of our manuscript and the positive assessment.

I have some questions and comments about the contents in the manuscript.

(i) Why does the energy spectrum which is the function of the wave number vector becomes discontinuous in Fig. 2? We can see in Fig. S4 that the discontinuity appears in the vicinity of the light cone. Does the system exhibit the phase transition around this region?

The discontinuity is the consequence of different optical responses of quasimomenta modes in an infinite system with long-range resonant dipole-dipole interaction (RDDI). As shown in section S1-1, the RDDI in the momentum space has a pole at $k_z^2 = (\omega_0/c)^2 - \mathbf{k}_{\parallel}^2$, where \mathbf{k}_{\parallel} is the Bloch momentum in the 2D irreducible Brillouin zone and ω_0 is the atomic transition frequency. For $(\omega_0/c)^2 > \mathbf{k}_{\parallel}^2$, k_z is real and the excitation within the light cone couples to the outgoing radiation and contributes to a nonzero decay rate. In contrast, for $(\omega_0/c)^2 < \mathbf{k}_{\parallel}^2$, k_z is imaginary and the excitation outside the light cone corresponds to an evanescent wave confined on the atomic array and does not contribute to decay. Hence, the complex eigenenergy changes to the real one as the Bloch momentum goes across the light cone from the center of 2D Brillouin zone. Therefore, it is not like a phase transition since the discontinuity arises not from tuning certain parameters of the system. It represents different excitation modes in the 2D Brillouin zone associated with distinct optical responses.

To clarify this property arising from RDDI, we have added a sentence on this in the “**Non-Hermitian degeneracy points**” section in the main text (underlined parts are modified from the previous version):

“According to the expression of RDDI in the reciprocal space (Supplementary section S1), the 2D Brillouin zone exhibits two kinds of distinct collective excitations separated by the light cone $|k| = 2\pi/\lambda$ [1] (FIG. 1a,b), wherein the dissipative modes couple to far-field radiation, while the modes with $|k| > 2\pi/\lambda$ related to evanescent wave confined to the atomic lattice plane are dissipationless.”

(ii) I think that Fig. 2 does not express the geometry-dependent non-Hermitian skin effect. This figure only shows that in the ribbon geometry which is regarded as a one-dimensional system, the conventional non-Hermitian skin effect occurs, and the energy spectrum corresponding the localized states forms the winding structure. However, Ref. [34] proposed the correspondence between the spectral area in a two-dimensional system with a full periodic boundary condition and the non-Hermitian skin effect which only occurs in a higher-dimensional system. Thus, the authors confused the discussion in a one-dimensional system with that in a two-dimensional system, and the discussion provided here is misleading. Therefore, the authors should modify this point.

We thank the Reviewer for raising this point. The ribbon geometry has the translational symmetry in one direction, but has open boundaries in the other direction, which could be perpendicular or oblique to the former one. Therefore, the emergence of non-Hermitian skin effect (NHSE) in the ribbon geometry depends on the orientation of open boundaries. That is why we claim the skin effect is geometry-dependent. In the Fig. 2 in the original manuscript, we just showed the NHSE for the boundaries open on the $(\bar{1}1)$ plane. In the revised manuscript, we add Fig. 2d-f for a rectangular atomic lattice with boundaries open in the y direction. This should signify the geometry dependence of NHSE in a ribbon geometry.

Following the advice of the referee, we have moved the discussion addressing arXiv:2102.05059 to the “**Non-Hermitian skin effect in 2D finite-size systems**” in the revised version (page 6). We elaborate more on this in our response to comment (vi).

(iii) I understand the mechanism of the occurrence of the scale-free localization in the ribbon geometry because of the long-range interaction. This phenomenon is interesting because it is associated with the non-Hermitian skin effect, and it has no counterpart in Hermitian systems. However, I think that it is difficult to make the ribbon geometry in the atomic-lattice system. Thus, how can we observe the scale-free localization in this system? Furthermore, does the scale-free localization occur under a full open boundary condition?

We agree that the ribbon geometry might be not easy to achieve in atomic-lattice systems since it extends infinitely along one direction. In the following we will provide an idea of how to observe the scale-free localization in a finite system and we will give numerical evidences.

We first consider a rectangular-lattice ribbon. It has a width of L_x unit cells and hosts skin modes at the boundaries in x-direction. Now let us shorten this ribbon along the infinite direction and impose open boundaries in this direction. If in this direction the number of unit cells, L_y , is large enough and the boundaries in this direction do not host skin modes, it is expected that such a finite system would behave similarly to a ribbon geometry and also exhibit the scale-free localization.

To numerically identify the scaling behavior of skin modes under a full open boundary condition, we fix $L_y = 70$ but vary L_x from 20 to 90 for a rectangular lattice with parallelogram-shaped boundary (See in the below plot). We start by choosing a skin mode at $L_x = 20$. At different L_x (from 30 to 90) we track the skin mode with eigenenergy closest to the energy of the chosen skin mode at $L_x = 20$. Next, we characterize the decay length of these skin modes along a 1D, staircase-like chain and do the scaling analysis. Different choices of 1D chains are shown in gray and blue in **b** below. The rescaled probability distributions in **d** and **e** reflect the scale-free localization, and the fitting results in **f** indicate that the results do not depend on the choice of the 1D chains.

Based on above discussion, in the section S5-2 in the revised supplementary information, we have added the details of analysis about the scaling behavior of the characteristic length in a finite 2D system.

Scaling behavior of the skin mode in a rectangular lattice with parallelogram-shaped boundary. **a**, Spectrum of a rectangular lattice ($a/\lambda = 0.3$, $\eta = 0.8$) with system size $L_x \times L_y = 80 \times 70$. The upper (lower) inset is a zoomed-in figure of the skin mode (blue) for $L_x = 20$ (80). **b**, Probability distributions of considered skin modes for $L_x = 80$. The staircase-like configurations of atoms (gray and blue) can be regarded as an effective 1D chain with $L_\perp = L_x$ unit cell. The sites along the 1D chain are labeled by r_\perp . **c**, Normalized probability distribution $|\psi(r_\perp)|^2$ and **d**, the rescaled probability $L_\perp |\psi(r_\perp/L_\perp)|^2$ of the gray 1D chain for different L_\perp . Darker (lighter) color corresponds to smaller (larger) system size L_\perp . The inset in **d** shows the fitting results for $L_\perp = 90$. **e**, The rescaled probability along the blue 1D chain. **f**, Scaling behavior of the characteristic length ξ_\perp . Gray (blue) gradient corresponds to the gray (blue) 1D chain in **b**.

(iv) In Fig. 4c, why does the eigenstates seem to be localized at the corners? Do these eigenstates exhibit the non-Hermitian skin effect?

The slight accumulation of eigenstates at the corners is because of the long-range interaction and these eigenstates at the corners do not exhibit the non-Hermitian skin effect. First of all, we numerically check the system-size scaling of the states at the corners in an $L \times L$ finite-size system with the same geometry shown in Fig. 4c. The results show that there is neither the higher-order non-Hermitian skin effect [Phys. Rev. Lett. 123, 016805 (2019)] with $\mathcal{O}(L^2)$ skin corner modes nor the second-order non-Hermitian skin effect [Phys. Rev. B 102, 205118 (2020)] with $\mathcal{O}(L)$ corner modes.

We further find the modes corresponding to the light cone contribute to the slight accumulation. Since the light cone arises from the long-range interaction, the long-range interaction should be responsible to the accumulation. In avoid of the confusion on this point, this is now clarified in “**Non-Hermitian skin effect in 2D finite-size systems.**” session of the main text (page 6):

“We note that the corner accumulation of $S_{NH}(r)$ in FIG. 4c does not show NHSE; instead, it arises from the long-range interaction (Supplementary FIG. S6).”

We also presented the details of analysis in section S5-1 in the revised supplementary information.

(v) In Figs. 4b and 4d, where are the nondefective degeneracy point and the exceptional point?

We thank the reviewer for raising this point. To clarify these nondefective degeneracy and exceptional points, in the revised main text, the nondefective degeneracy point in Fig. 4a, c and the exceptional point in Fig. 4b, d are denoted as the red circles and red rectangles, respectively.

(vi) In this work, the authors only focus on the rectangle geometry and on the parallelogram geometry. I think that the non-Hermitian skin effect also occurs in other geometry. How can we quantitatively investigate the dependence on geometry of the non-Hermitian skin effect?

We agree that the non-Hermitian skin effect can occur in other geometry. Since there are infinite possibilities of boundaries and geometric shapes for two-dimensional finite-size systems, it is challenging to derive a general criterion which determine the emergence of non-Hermitian skin effect. Therefore, for a given geometry, numerical calculation is still necessary. In addition to that, in our paper we try to provide an idea to determine if there are skin modes localized on an open boundary of a finite system: We can extend the system infinitely along this open boundary to reconstruct a ribbon geometry. Once the bulk spectrum of this ribbon geometry can form a loop, it indicates the emergence of skin modes on this boundary. This idea provides a way to

understand the geometry-dependent NHSE, which is now addressed on page 6 in “**Non-Hermitian skin effect in 2D finite-size systems**” session of the main text (underlined parts are modified from the previous version):

“To understand the NHSE here, we can extend a finite system infinitely along one open boundary to reconstruct a ribbon geometry. This allows us to determine which boundary extensive skin modes are localized at by identifying its energy spectrum topology. For instance, there are two possible ribbon geometries for the finite system in FIG. 4d: one extends infinitely along the oblique direction (FIG. 2a) and the other extends infinitely along the horizontal direction (FIG. 2d). The skin modes emerge in the former case (FIG. 2b), while the NHSE in the latter case is suppressed by the mirror symmetry (FIG. 2e). Therefore, the population distributions of non-Hermitian eigenstates rely on the orientations of open boundaries and the lattice configurations. The comparison with the ribbon geometry provides a way to understand a so-called geometry-dependent NHSE [40] in two and higher dimensional systems. In the same manner, we note that the scale-free localization found in a ribbon geometry could also be observed in a 2D finite-size lattice (supplementary FIG. S7).”

If the authors answer my questions and appropriately modify the manuscript, I recommend to publish this paper in Nature Communications.

We thank the Reviewer for these constructive questions and comments, which help us further improve our manuscript. We hope that our revised version is satisfactory for publication in Nature Communications.

REVIEWERS' COMMENTS

Reviewer #1 (Remarks to the Author):

The authors have provided detailed comments and replies to the previous round of refereeing. I disagree with the comment that the method of calculating the band structure in References [29,30] involves an artificial cutoff. This is a rigorous method of evaluating the band structure and alternative to Ewald summation used e.g. in Phys. Rev. A 96, 041603(R) (2017). The theory is non-relativistic, so it necessary has $1/r^3$ Lamb shift divergence.

Apart this, the work is valuable and interesting. I would, however, be more comfortable recommending publication in Nature Communications if the authors can better highlight (and better focus on) in the abstract and introduction the important and novel aspects of their findings as in the rebuttal letters.

Reviewer #2 (Remarks to the Author):

I think that the authors appropriately answered my questions. Furthermore, they modified the unclear parts, e.g., description of the geometry-dependent non-Hermitian skin effect and the scale-free localization, in the main text and the supplementary material. In particular, it is obvious that we can see the appearance of the geometry-dependent non-Hermitian skin effect from Fig. 2.

We expect that the present work paves a way to study the non-Hermitian skin effect in a higher-dimensional system in terms of both theories and experiments. Therefore, I would like to recommend to publish this paper in Nature Communications.

Reviewer #1 (Remarks to the Author):

The authors have provided detailed comments and replies to the previous round of refereeing. I disagree with the comment that the method of calculating the band structure in References [29,30] involves an artificial cutoff. This is a rigorous method of evaluating the band structure and alternative to Ewald summation used e.g. in Phys. Rev. A 96, 041603(R) (2017). The theory is non-relativistic, so it necessary has $1/r^3$ Lamb shift divergence.

We agree that the method requiring frequency cutoff is not an artificial approach. Since we only discussed this point in the previous response letter, we would not elaborate more on this in the manuscript in this round of revision. Here we would like to clarify again the difference between our method and those used in Ref. [29,30]: we are dealing with the collective Lamb shift, $\sum_{R \neq 0} e^{-ik \cdot R} G_0(R)$, which is convergent and independent of the frequency cutoff. Unlike requiring a specific cutoff in Ref. [29,30], our Euler-Maclaurin formula does not require any frequency cutoff.

Apart this, the work is valuable and interesting. I would, however, be more comfortable recommending publication in Nature Communications if the authors can better highlight (and better focus on) in the abstract and introduction the important and novel aspects of their findings as in the rebuttal letters.

We thank the Reviewer again for appreciation of our work. Following the Reviewer's suggestions on our abstract and the introduction, we have modified them by emphasizing the correspondence among the exceptional point, nontrivial winding in the complex energy plane, and the non-Hermitian skin effect in a ribbon geometry (underlined parts are modified from the previous version):

The abstract

"Here we show that exceptional points develop from a nondefective degeneracy by lowering the crystal symmetry of a square atomic lattice, and dispersive bulk Fermi arcs that originate from exceptional points are truncated by the light cone. From its nontrivial energy spectra topology, we demonstrate that the geometry-dependent non-Hermitian skin effect emerges in a ribbon geometry."

The last paragraph in Introduction

"Due to the inherent non-Hermiticity of dipole-dipole interaction, there is no Hermitian limit of our system, such that EPs should emerge from the mechanism different from splitting Hermitian degeneracy points by adding non-Hermiticity. Here we demonstrate that paired EPs can be split from a symmetry-protected nondefective degeneracy point (NDP) by a symmetry-breaking perturbation. With the nontrivial winding in the complex energy plane arising from EPs, we find that a ribbon geometry exhibits extensive geometry-dependent skin modes."

Reviewer #2 (Remarks to the Author):

I think that the authors appropriately answered my questions. Furthermore, they modified the unclear parts, e.g., description of the geometry-dependent non-Hermitian skin effect and the scale-free localization, in the main text and the supplementary material. In particular, it is obvious that we can see the appearance of the geometry-dependent non-Hermitian skin effect from Fig. 2.

We expect that the present work paves a way to study the non-Hermitian skin effect in a higher-dimensional system in terms of both theories and experiments. Therefore, I would like to recommend to publish this paper in Nature Communications.

We are thankful to the Reviewer for these valuable suggestions and comments raised in the previous round of reviewing, and we thank the Reviewer again for the recommendation of our manuscript.